# UNDERSTANDING & GENERALIZING ALPHAGO ZERO

## ABSTRACT

AlphaGo Zero (AGZ) (Silver et al., 2017b) introduced a new *tabula rasa* reinforcement learning algorithm that has achieved superhuman performance in the games of Go, Chess, and Shogi with no prior knowledge other than the rules of the game. This success naturally begs the question whether it is possible to develop similar high-performance reinforcement learning algorithms for generic sequential decision-making problems (beyond two-player games), using only the constraints of the environment as the "rules." To address this challenge, we start by taking steps towards developing a formal understanding of AGZ. AGZ includes two key innovations: (1) it learns a policy (represented as a neural network) using *supervised learning* with cross-entropy loss from samples generated via Monte-Carlo Tree Search (MCTS); (2) it uses *self-play* to learn without training data.
We argue that the self-play in AGZ corresponds to learning a Nash equilibrium for the two-player game; and the supervised learning with MCTS is attempting to learn the policy corresponding to the Nash equilibrium, by establishing a novel bound on the difference between the expected return achieved by two policies in terms of the expected KL divergence (cross-entropy) of their induced distributions. To extend AGZ to generic sequential decision-making problems, we introduce a *robust MDP* framework, in which the agent and nature effectively play a zero-sum game: the agent aims to take actions to maximize reward while nature seeks state transitions, subject to the constraints of that environment, that minimize the agent's reward. For a challenging network scheduling domain, we find that AGZ within the robust MDP framework provides near-optimal performance, matching one of the best known scheduling policies that has taken the networking community three decades of intensive research to develop.

## 1 INTRODUCTION

In 2016, *AlphaGo* (Silver et al., 2016) became the first program to defeat the world champion in the game of Go. Soon after, another program, *AlphaGo Zero* (AGZ) (Silver et al., 2017b), achieved even stronger performance despite learning the game from scratch given only the rules. Starting *tabula rasa*, AGZ mastered the game of Go entirely through self-play using a new reinforcement learning algorithm. The same algorithm was recently shown to achieve superhuman performance in Chess and Shogi (Silver et al., 2017a). The goal of this paper is to develop a formal understanding of AGZ's learning algorithm, and to extend it to sequential decision problems beyond two-player games.

**Overview of AGZ.** AGZ is designed for a two-player *zero-sum* game such as Go. The core idea of AGZ is to generate training data from games of self-play, and use this data to train the policy via supervised learning. Specifically, AGZ uses a combined policy and value network architecture, i.e., a single neural network $g_\theta$ with parameters $\theta$. The input to the network is the state $s$, which encodes both the current player *and* the state of the game (e.g. board configuration in Go). The network then outputs $g_\theta(s) = (\pi_\theta, v_\theta)$, where the vector $\pi_\theta(\cdot|s)$ represents a probability distribution over the possible moves, and $v_\theta$ is a scalar value representing the probability of the current player winning.

To generate the training data, AGZ simulates the game multiple times (starting with empty board in Go). During each simulation, AGZ plays the game against itself. That is, for each player's turn, it uses the current policy network $g_\theta$ aided by a Monte-Carlo tree search (MCTS) (Kocsis & Szepesvári, 2006; Kocsis et al., 2006; Browne et al., 2012) to generate the next move. Specifically, for the current player and state, MCTS outputs a distribution $\pi(\cdot|s)$ over all possible actions, and AGZ picks an action by randomly sampling from this distribution. This generates a training data point $(s, \pi(\cdot|s), z)$,

where $z$ is a reward assigned retroactively based on whether the player taking the action won (+1) or lost (-1) that particular game.

With the training data obtained as above, the neural network parameters $\theta$ are continuously updated with a mini-batch of data $M$, sampled uniformly among all time-steps of the most recent game(s) of self-play. In particular, the parameters $\theta$ are optimized by stochastic gradient descent on the following $\ell_2$-regularized loss function that sums over mean-squared error and cross-entropy loss respectively,

$$\text{loss} = (z - v_\theta)^2 - \pi(\cdot|s) \log \pi_\theta(\cdot|s) + \lambda \|\theta\|^2, \tag{1}$$

where the quantity $\pi(\cdot|s) \log \pi_\theta(\cdot|s)$ represents the cross-entropy or KL divergence between the two distributions. The pseudocode describing the above algorithm is provided in Appendix A.

In (Silver et al., 2017b), authors refer to MCTS as a "strong policy improvement operator". The intuition is that, starting with the current policy, MCTS produces a better target policy for the current state. Thus training the neural network to minimize cross entropy loss with respect to the target policy on the current state is a form of policy iteration (Sutton & Barto, 1998), which will hopefully improve the neural network policy. Although this explanation seems natural, and is intuitively appealing, it is not at all clear why this MCTS-driven policy iteration should work.

**The questions.** We are interested in three key unanswered questions about AGZ:

> *Question 1. What is the optimal policy that AGZ is trying to learn?*

We would like a formal model to understand how learning via self-play impacts the policy to which AGZ converges.

> *Question 2. Why is minimizing cross-entropy (or KL divergence) the right objective?*

We seek to precisely characterize the distribution under which cross-entropy should be evaluated, and derive quantitative performance guarantees that explain what optimizing for this metric achieves.

> *Question 3. How does AGZ extend to generic sequential decision-making problems?*

In generic sequential decision-making, the agent chooses actions and earns rewards while the state evolves per the environment. This is different from two-player zero-sum games for which AGZ was designed.

**Summary of results.** As the main contribution of this work, we answer all of the above questions.

*AGZ's optimal policy is a Nash equilibrium.* We study AGZ's learning algorithm under the formalism of two-player Markov games. The notion of self-play naturally defines the *optimal* policy as a *Nash equilibrium* for the two-player zero-sum game. That is, AGZ is attempting to learn this Nash equilibrium. See Section 2 and Section 4 for the details.

*KL divergence bounds distance to optimal reward.* AGZ uses the cross-entropy (or KL divergence) metric as a loss function for the supervised learning. To justify this theoretically, we develop a novel bound on the difference between the reward associated with two policies in terms of the cross-entropy distance between them. Specifically, let $\pi = (\pi_1, \pi_2)$ denote game-playing policy, where $\pi_1$ and $\pi_2$ are the policies of the respective players (e.g. the Black player and the White player in Go). We bound the difference between the cumulative discounted reward (i.e., the "return") achieved with an arbitrary policy $\pi(\cdot|s)$ compared to the optimal policy $\pi^*(\cdot|s)$ (i.e., Nash equilibrium) in terms of the expected KL divergence between $\pi(\cdot|s)$ and $\pi^*(\cdot|s)$. The expectation is with respect to states sampled according to rollouts with the optimal policy $\pi^*$. The bound has the following form:

$$\left| R(\pi) - R(\pi^*) \right| \leq C \sqrt{\mathbb{E}_{s \sim \overline{\rho}_{\pi^*}} \left[ D_{\text{KL}}\big(\pi^*_{I(s)}(\cdot|s) \| \pi_{I(s)}(\cdot|s)\big) \right]}, \tag{2}$$

where $R(\pi)$ and $R(\pi^*)$ are the returns for policies $\pi$ and $\pi^*$ respectively, $C$ is a constant, $I(s)$ denotes the player to play at state $s$, and $\overline{\rho}_{\pi^*}(s)$ is the appropriately normalized state visit frequencies when actions are taken according to policy $\pi^*$. See Section 3 for precise definitions and details. In the process, we establish an invariant for Markov games that is equivalent to the result by (Kakade & Langford, 2002) established for the MDP setting (see Appendix D).

This result provides an immediate justification for AGZ's core learning algorithm. Notice that if the expected KL divergence between a given policy and the optimal policy (i.e. Nash equilibrium) goes

to 0, then the return for that policy must approach that of the optimal policy. Therefore, if MCTS produces reasonable estimates of the optimal policy for the states that are sampled, then supervised learning with cross entropy loss for states sampled via the MCTS target policy should ideally converge to the desired optimal policy (i.e. Nash equilibrium). Interestingly enough, prior work has shown that MCTS asymptotically converges to the optimal actions in both two-player zero-sum games (Kocsis et al., 2006) as well as discounted MDPs (Chang et al., 2005; Kocsis & Szepesvári, 2006).

*Robust MDP as a two-player game.* To extend AGZ beyond the two-player game setting, we introduce a robust MDP framework. As in a standard MDP, the agent chooses an action given the current state to earn a reward, and the state makes a transition according to the environment. In a traditional MDP, state transitions occur as per an (un)known *fixed* Markov kernel. In the robust MDP, the transition of state is decided by "nature," a separate decision maker that selects state transitions subject to *environmental constraints*. The goal of the agent is to choose actions to maximize the long-term reward, while the goal of nature is to choose state transitions (subject to environmental constraints) to minimize the long-term reward earned by the agent. In effect, robust MDP is a two-player *zero-sum* game between the agent and nature. By transforming generic sequential decision-making to a two-player game, we can apply AGZ to a much broader class of problems. We note that a similar adversarial view has been explored recently to model reinforcement learning as simultaneous games (Pinto et al., 2017). Our theoretical results provide new insights about such adversarial approaches in both turn-based and simultaneous game models.

*Empirical results.* Is the robust MDP framework useful in practice? To answer this question, we consider the challenging problem of learning a scheduling algorithm for an input-queued switch, see McKeown (1999); McKeown et al. (1999); Giaccone et al. (2002); Shah & Wischik (2006); Shah et al. (2012); Maguluri & Srikant (2016). Like a traffic junction, the role of an input-queued switch is to schedule the transfer of incoming packets to outgoing ports, e.g. across a network switch or a data center. In an input-queued switch with $n$ incoming/outgoing ports, scheduling involves selecting one of the $n!$ possible matchings between inputs and outputs. The goal is to transfer as many packets as possible across the switch while minimizing the packet delay. Scheduling algorithms for input-queued switches have been the subject of intensive research effort in the networking community, including attempts using approximate dynamic programming Moallemi et al. (2008). This is because on one hand, input-queued switch is like the *E-coli* of network scheduling; progress on scheduling input-queued switches has provided many insights about more general network scheduling problems. On the other hand, the problem is of great importance in its own right in practice.

We utilize the framework of robust MDP along with the AGZ algorithm to find a scheduling policy that attempts to minimize the average packet delay for any admissible packet arrival process to the switch. The resulting policy performs nearly as well as the best known policy in the literature (cf. Shah & Wischik (2006); Shah et al. (2012); Maguluri & Srikant (2016)) and provides performance that is close to the fundamental lower bound. This is remarkable especially given that an equivalent policy was discovered after an intensive effort by experts spanning over three decades of research. See Section 5 for details.

In summary, this paper serves as a first attempt towards a principled understanding of AGZ under minimal assumptions. Given the large body of prior work in the context of decision making, we provided a brief overview of the most closely related previous work, and defer additional discussion of related work to Appendix B.

## 2    TWO-PLAYER GAME, ROBUST MDP & NASH EQUILIBRIUM

We introduce a formal model to capture the essence of AlphaGo Zero (AGZ). In particular, we introduce the setting of two-player zero-sum game for which AGZ is primarily designed. Next, we shall explain how a generic MDP can be reduced to the setting of the two-player zero-sum game. For the setting of two-player game, the ideal solution is given by the Nash Equilibrium. We posit that AGZ, in effect, is trying to learn this Nash equilibrium. The transformation of generic MDP to the setting of two-player zero-sum game allows us to extend AGZ for the setup of generic decision making. Therefore, it is sufficient to study the performance of AGZ for the setting of two-player zero-sum game to understand it's efficacy for generic setting as well.

**Two-player zero-sum game.** Most games, including Go, exhibit a "turn-based" feature: players take turns to make decisions. To capture this feature, we consider a two-player *turn-based* Markov game, where the two players alternatively take an action. Note that in general, turn-based Markov games can be viewed as a special class of stochastic games (a.k.a. Markov games) by assigning a dummy action, in each step, to the player who does not play in that step. We retain the turn-based formulation in the main text, since it is more closely aligned with the Go setting. Extension to the general Markov games is straightforward and is discussed in detail in Appendix E.

For simplicity, we denote the two players, player 1 and player 2, by P1 and P2. This game can be expressed as the tuple $(\mathcal{S}_1, \mathcal{S}_2, \mathcal{A}_1, \mathcal{A}_2, r, P, \gamma)$. $\mathcal{S}_1$ and $\mathcal{S}_2$ are the set of states controlled by P1 and P2, respectively. For example, in Go, the two sets could be the board configurations whose current players are the Black player and the White player, respectively. Accordingly, $\mathcal{A}_i(s)$ is the set of feasible actions for player $i$ in any given state $s \in \mathcal{S}_i$. For turn-based games such as Go, it is typical to assume that $\mathcal{S}_1 \cap \mathcal{S}_2 = \emptyset$ [1]. Let us denote by $\mathcal{S} = \mathcal{S}_1 \cup \mathcal{S}_2$ the entire state space of the game.

For each state $s \in \mathcal{S}$, let $I(s) \in \{1, 2\}$ indicate the current player to play. At state $s$, upon taking action $a$ by the corresponding player $I(s)$, player $i \in \{1, 2\}$ receives a reward $r^i(s, a)$. In most two-player games such as Go and Chess, the two players play against each other, and the reward a player receives is equal to the negative reward (i.e., loss) of the other player. To reflect such structures, we henceforth focus on two-player zero-sum games where $r^1(s, a) = -r^2(s, a)$. Without loss of generality, we let P1 be our reference and use the notation $r(s, a) \triangleq r^1(s, a)$ for later definitions of value functions. The reward function is assumed to be uniformly bounded by some constant $C_R$. In addition, let $P(s, a)$ be the distribution of the new state after playing action $a$, in state $s$, by player $I(s)$. Note that in Go/Chess, the state transitions are deterministic.

For each $i \in \{1, 2\}$, let $\pi_i$ be the policy for player $i$, where $\pi_i(\cdot|s)$ is a probability distribution over $\mathcal{A}_i(s)$. Denote by $\Pi_i$ the set of all policies of player $i$, and let $\Pi = \Pi_1 \times \Pi_2$ be the set of all polices for the game. Given a policy $\pi = (\pi_1, \pi_2)$, the total discounted reward $R(\pi)$ is given by

$$R(\pi) = \mathbb{E}_{s_0 \sim \rho_0, a_t \sim \pi_{I(s_t)}(\cdot|s_t), s_{t+1} \sim P(s_t, a_t)} \left[ \sum_{t=0}^{\infty} \gamma^t r(s_t, a_t) \right], \tag{3}$$

where $\rho_0 : \mathcal{S} \to [0, 1]$ is the distribution of the initial state $s_0$, and $\gamma \in (0, 1)$ is the discount factor. We use the following standard definition of the value function in games:

$$V_{\pi_1, \pi_2}(s) = \mathbb{E}_{a_t, s_{t+1}, a_{t+1}, \dots} \left[ \sum_{k=0}^{\infty} \gamma^k r(s_{t+k}, a_{t+k}) | s_t = s \right],$$

where $a_l \sim \pi_{I(s_l)}(\cdot|s_l)$ and $s_{l+1} \sim P(s_l, a_l)$. That is, $V_{\pi_1, \pi_2}(s)$ is the expected total discounted reward for P1 if the game starts from state $s$, players P1 and P2 respectively use the policies $\pi_1$ and $\pi_2$. A two-player zero-sum game can be seen as player P1 aiming to maximize the accumulated discounted reward while P2 attempting to minimize it.

**Robust MDP as a two-player zero-sum game.** We introduce setting of "robust" MDP to extend AGZ for generic decision making. Typically, the generic sequential decision making framework is captured through tuple $(\mathcal{S}, \mathcal{A}, r, \gamma, P)$: $\mathcal{S}$ is the set of all states, $\mathcal{A}$ is the set of all actions agent can take, $r : \mathcal{S} \times \mathcal{A} \to \mathbb{R}$ denotes the reward function, $\gamma \in (0, 1)$ is discount factor, and for each $(s, a) \in \mathcal{S} \times \mathcal{A}$, $P_{s,a} : \mathcal{S} \to [0, 1]$ denotes the Markov transition kernel with $P_{s,a}(s')$ being the probability of transitioning into state $s' \in \mathcal{S}$ from state $s$ given action $a$. At each time instance $t$, given state $s_t \in \mathcal{S}$, agent takes action $a_t$ which leads to reward $r(s_t, a_t)$ and the nature makes the state transition to $s_{t+1} \in \mathcal{S}$ as per the Markov transition kernel $P_{s_t, a_t}$. Given such a setting, traditionally there are two popular approaches. One, the Markov Decision Process (MDP), where one assumes the prior knowledge of nature, i.e. $P$, and solves for optimal policy for the agent defined through Bellman's equations and dynamic programming. This includes approaches such as policy iteration, value iteration or its variants, cf. Bertsekas et al. (2005); Bertsekas & Tsitsiklis (1995). Second, the reinforcement learning (RL) where $P$ is assumed to be unknown but aspects of it are learnt along-side of the optimal policy using approximate dynamic programming such as Q-learning.

---

[1]This is a reasonable assumption in general in any setting when only one player is taking a "turn" to play, since one may incorporate the "turn" of the player as part of the state, and thus by definition a player 1's state space (i.e. when it is player 1's turn) is disjoint from that of player 2's turn.

In contrast, here we model nature as an *adversary*. Precisely, we model the decision making as a game between agent and nature. Agent is P1 and nature is P2. The agent is trying to chose action with the aim of maximizing reward while nature is trying to make state transition so that agent's reward is minimized. Of course, if nature can make arbitrary transitions then it will result into a degenerate setting where nature will always set state to the one that leads to minimal reward. In reality, there are constraints per which state can transition. Specifically, let $\mathcal{C}(s, a) \subset \mathcal{S}$ be the set of all allowable states the transition is feasible. And nature tries to choose action $a' \in \mathcal{A}_2(s, a)$ that leads to state $s' \in \mathcal{C}(s, a)$ to minimize (future) reward of the agent.

Now, we can view this as a two-player zero-sum game. The player P1 is agent with possible actions $\mathcal{A}_1 = \mathcal{A}$ with states $\mathcal{S}_1 = \mathcal{S}$, and denote by $f(s, a)$ the state after selecting action $a$ in state $s$. The player P2 is the nature with states $\mathcal{S}_2 = \{f(s, a) | s \in \mathcal{S}, a \in \mathcal{A}\}$ and actions $\mathcal{A}_2 = \cup_{s \in \mathcal{S}, a \in \mathcal{A}} \mathcal{A}_2(s, a)$. When player P1 takes action $a \in \mathcal{A}$ given state $s \in \mathcal{S}$, the system transition to intermediate state $f(s, a)$ and reward generated for P1 is $r^1(s, a)$. The player P2 receives reward $r^2(s, a) = -r^1(s, a)$. Then given state $f(s, a)$, P2 takes action $a' \in \mathcal{A}_2(s, a)$, the state transitions to $s' \in \mathcal{S}_1$, reward generated for player P2 is $r^2(f(s, a), a')$ and reward for P1 is $r^1(f(s, a), a') = -r^2(f(s, a), a')$. Let $r(s, a) \triangleq r^1(s, a) + r^1(f(s, a), a')$. We denote by $s_t$ the states that are observed and $a_t \in \mathcal{A}_1$ be the actions taken by player P1 at time $t$. Then the cumulative discounted reward earned by player P1 is precisely $\sum_{t=0}^{\infty} \gamma^t r(s_t, a_t)$. Thus, reward of player P1 precisely matches that of the original decision making agent and the setting of decision making is precisely converted into a zero-sum game between two players, agent and nature.

Given the above transformation, in the remainder of this work, we will focus on two-player zero-sum game with interest in both the classical turn-based game as well as the robust MDP setup described. The analysis of AGZ in the two-player zero-sum game naturally will apply to the generic decision making setting through the above described transformation.

**Nash equilibrium.** In the game setting, we are interested in the minimax equilibrium solutions. To describe this concept, we view the value function $V_{\pi_1, \pi_2}$ as a vector in $\mathbb{R}^{|\mathcal{S}|}$. Given two vectors $U$ and $W$, we use the notation $U \leq W$ to indicate element-wise comparisons. Recall that P1 is the reward maximizer while P2 is the minimizer.

**Definition 1.** *(Optimal Counter Policy) Fix a policy $\pi_2 \in \Pi_2$ for* P2. *A policy $\pi_1$ for* P1 *is said to be an optimal counter-policy against $\pi_2$, if and only if*

$$V_{\pi_1, \pi_2} \geq V_{\pi_1', \pi_2}, \forall \pi_1' \in \Pi_1.$$

*Similarly, a policy $\pi_2 \in \Pi_2$ for* P2 *is said to be an optimal counter-policy against a fix policy $\pi_1 \in \Pi_1$ for* P1*, if and only if*

$$V_{\pi_1, \pi_2} \leq V_{\pi_1, \pi_2'}, \forall \pi_2' \in \Pi_2.$$

In a two-player zero-sum game, it has been shown that the pairs of optimal policies coincide with the Nash equilibrium of this game (Patek, 1997; Hansen et al., 2013; Perolat et al., 2015). In particular, a pair of policies $(\pi_1^*, \pi_2^*)$ is called an equilibrium solution of the game, if $\pi_1^*$ is an optimal counter policy against $\pi_2^*$ and $\pi_2^*$ is an optimal counter policy against $\pi_1^*$. The value function of the optimal policy, $V_{\pi_1^*, \pi_2^*}$, is the *unique* fixed point of a $\gamma$-contraction operator Hansen et al. (2013). Consequently, the value of the equilibrium solution, $R(\pi_1^*, \pi_2^*) = \mathbb{E}_{s_0 \sim \rho_0}[V_{\pi_1^*, \pi_2^*}(s_0)]$ (cf. Equation (3)) is unique. In the sequel, we will simply refer to the strategy $\pi^* = (\pi_1^*, \pi_2^*)$ as the optimal policy. Using our key result Theorem 2, in Section 4, we shall argue that AGZ in effect is trying to learn this Nash equilibrium.

## 3 Key Theorem

As our key result, we establish a novel bound on the difference of the expected rewards of any policy and that of the optimal one (i.e., Nash equilibrium) in terms of the KL divergence (i.e., cross entropy) of these two policies. This will play a crucial role in justifying AGZ trying to learning the Nash equilibrium as argued in Section 4.

Define $\rho_\pi(s)$ as the unnormalized discounted visit frequencies for state $s$ under policy $\pi = (\pi_1, \pi_2)$:

$$\rho_\pi(s) = P(s_0 = s) + \gamma P(s_1 = s) + \gamma^2 P(s_2 = s) + \dots, \tag{4}$$

where $s_0 \sim \rho_0$ and actions $a_t$ are taken according to policy $\pi_{I(s_t)}$. Note that $\sum_{s \in \mathcal{S}} \rho_\pi(s) = 1/(1-\gamma)$; let $\overline{\rho}_\pi = (1-\gamma)\rho_\pi$ be the normalized probability distribution. It is easy to see that as $\gamma \to 1$, $\overline{\rho}_\pi$ approaches $\mu_\pi$, the stationary distribution induced by the policy $\pi$; in particular, $\overline{\rho}_{\pi^*}$ approaches $\mu^*$ for large enough discount factor.

**Theorem 2.** *Given a two-player zero-sum game, for any policy $\pi = (\pi_1, \pi_2)$ and Nash equilibrium $\pi^* = (\pi_1^*, \pi_2^*)$,*

$$|R(\pi) - R(\pi^*)| \leq C \sqrt{\mathbb{E}_{s \sim \bar{\rho}_{\pi^*}} \left[ D_{KL}\big(\pi_{I(s)}^*(\cdot|s) \| \pi_{I(s)}(\cdot|s)\big) \right]}, \tag{5}$$

*where $C$ is a constant depending only on $\gamma$ and $C_R$.*

We remark that the above bound in fact holds for any pair of policies $\pi$ and $\tilde{\pi}$. The inequality (5) corresponds to taking $\tilde{\pi} = \pi^*$. The proof of Theorem 2 is provided in Appendix D.

## 4 ALPHAGO ZERO LEARNS NASH EQUILIBRIUM

Using Theorem 2, we shall argue that AGZ tries to approximate an ideal policy for learning the Nash equilibrium. To that end, we first describe this ideal policy suggested by Theorem 2 and then we discuss how AGZ approximates it.

### 4.1 IDEAL POLICY FOR LEARNING NASH EQUILIBRIUM

Consider a parametrized class of policies $\{(\pi_{\theta_1}, \pi_{\theta_2}) | \theta_1 \in \mathbb{R}^{m_1}, \theta_2 \in \mathbb{R}^{m_2}\}$ such that there exists $\theta^* = (\theta_1^*, \theta_2^*)$ so that $(\pi_{\theta_1^*}, \pi_{\theta_2^*}) = (\pi_1^*, \pi_2^*)$, the desired Nash equilibrium. We shall assume that such a $\theta^*$ is unique that we wish to find. Theorem 2 suggests that such $\theta^*$ can be found by minimizing the right-hand side of (5). To that end, define

$$f(\theta) = \mathbb{E}_{s \sim \bar{\rho}_{\pi^*}} \left[ D_{\text{KL}}\big(\pi_{I(s)}^*(\cdot|s) \| \pi_{\theta_{I(s)}}(\cdot|s)\big) \right]. \tag{6}$$

The gradient algorithm for finding $\theta^*$ would suggest iteratively update $\theta$ as

$$\theta \leftarrow \theta - \alpha \nabla_\theta f(\theta). \tag{7}$$

Assuming $f$ does not have spurious stationary points, the algorithm (with appropriately decaying constants $\alpha$) will converge to $\theta^*$.

This is an ideal policy. Because it requires calculating $\nabla_\theta f(\theta)$ which involves computing expectation with respect to $\rho_{\bar{\pi}^*}$, the normalized stationary distribution with respect to optimal policy. Next, we argue that AGZ implements stochastic approximation to (7).

### 4.2 AGZ'S APPROXIMATIONS OF IDEAL POLICY

AGZ implements idealized policy in (7) through a sequence of approximations that we describe next. As a consequence, if all the approximations work out, then AGZ would end up finding the optimal policy, i.e the Nash equilibrium.

*Optimal policy oracle via MCTS.* To evaluate $\nabla_\theta f(\theta)$ or $f(\theta)$, it is sufficient to have access to an oracle that given a state $s$, it provides $\pi_{I(s)}^*(\cdot|s)$. Because, once we have access to such an oracle, starting with $s_0 \sim \rho_0$, we can sample states $s_t$, $t \geq 1$ iteratively as follows: given $s_t$, obtain action $a_t \sim \pi_{I(s_t)}^*(\cdot|s_t)$, observe a new state $s_{t+1}$. Then, with respect to randomness over sampling of $a_t$ and resulting randomness in state transitions

$$\mathbb{E}\Big[ \sum_{t \geq 0} D_{\text{KL}}\big(\pi_{I(s_t)}^*(\cdot|s) \| \pi_{\theta_{I(s_t)}}(\cdot|s)\big) \Big] \propto f(\theta), \text{ and}$$

$$\mathbb{E}\Big[ \sum_{t \geq 0} \nabla_\theta D_{\text{KL}}\big(\pi_{I(s_t)}^*(\cdot|s) \| \pi_{\theta_{I(s_t)}}(\cdot|s)\big) \Big] \propto \nabla_\theta f(\theta). \tag{8}$$

Therefore, with access to the oracle described, stochastic approximation of (7) can be implemented. The pseudocode of this ideal algorithm, together with addition discussions on the form of the updates, may be found in Appendix F.

In principle, any approximate oracle with reasonable theoretical guarantees can be applied to replace the optimal oracle in practice. AGZ uses Monte Carlo Tree Search (MCTS) to obtain such an approximate oracle (Chang et al., 2005; Kocsis & Szepesvári, 2006; Kocsis et al., 2006). MCTS has many variants; the basic version, known as UCT (UCB applied to Trees), uses ideas from UCB (Upper Confidence Bounds) algorithm to guide the tree search. For this algorithm, it has been shown that the probability of taking suboptimal actions converges to zero (Kocsis & Szepesvári, 2006), and hence, the empirical action distribution eventually converges to the optimal action distribution $\pi^*(\cdot|s)$. We also note that in AGZ's network, it has a value function component as its output. This value function is only used during the MCTS phase to guide the search (in particular, to evaluate the newly expanded nodes), rather than directly applied to select actions. That is, we could view this UCT style search, aided by the value function, as a variant of MCTS with the hope of obtaining a more efficient approximate oracle than the vanilla MCTS.

*Stochastic gradient descent.* A naive stochastic approximation to (7) as mentioned above may obtain many samples of the random variable on the left-hand side in (8). However, that may require a lot more simulations and could be wasteful. Alternatively, the Stochastic Gradient Descent (SGD) would suggest making bolder updates: compute and update the gradient continually. And this is precisely what AGZ implements.

*Existence of and convergence to $\theta^*$.* For the ideal gradient algorithm (7) to be able to find optimal policy, there are two key assumptions made: (1) there exists $\theta^*$ so that $\pi^* = \pi_{\theta^*}$, and (2) the gradient algorithm finds such a $\theta^*$ starting from any initial parameter $\theta$. The implicit assumption made in AGZ implementation is that the policy neural network utilized is flexible enough that (1) is expected to satisfy. The assumption (2) is the mystery behind the success of any generic neural network based solution. And AGZ(-like) implementation hopes that (2) works out through various cleverness in terms of architecture choice and learning algorithm.

# 5 AGZ FOR ROBUST MDP – A CASE-STUDY

AGZ has been successful in the context of two-player games. The robust MDP formulation makes it applicable to the setting of generic sequential decision making. To understand how well AGZ does when applied to robust MDP, we consider a challenging networking problem as a case-study: scheduling transfer of packets in an input-queued switch.

**Background.** A router or a switch in Internet or a data center has several input ports and output ports. A data transmission cable is attached to each of these ports. Packets arrive at the input ports. The function of the switch is to work out which output port each packet should go to, and to transfer packets to the correct output ports. This last function is called *switching*. There are a number of possible architectures; we will consider the commercially popular input-queued switch architecture.

The switch operates in discrete time. At each time slot, the switch fabric can transmit a number of packets from input ports to output ports, subject to the two constraints that each input can transmit at most one packet, and that each output can receive at most one packet. In other words, at each time slot the switch can choose a *matching* from inputs to outputs. In general, for an $n$-port switch, there are $N = n^2$ queues. Packets arriving at input $k$ destined for output $\ell$ are stored at input port $k$, in queue $Q_{k,\ell}$. The schedule $\sigma \in \mathbb{R}_+^{n \times n}$ is given by $\sigma_{k,\ell} = 1$ if input port $k$ is matched to output port $\ell$ in a given time slot, and $\sigma_{k,\ell} = 0$ otherwise. By the matching constraints, the corresponding schedule set $\mathcal{S}$ is defined to be $\left\{ \sigma \in \{0,1\}^{n \times n} : \sum_{m=1}^n \sigma_{k,m} \leq 1, \sum_{m=1}^n \sigma_{m,\ell} \leq 1, 1 \leq k, \ell \leq n \right\}$. Suppose data packets arrive at queue $(k, \ell)$ at rate $\lambda_{k,\ell}$ for $1 \leq k, \ell \leq n$. Then, the effective load induced on the system, $\rho(\lambda)$, is given by $\rho(\lambda) = \max_{1 \leq k, \ell \leq n} \left\{ \sum_{m=1}^n \lambda_{k,m}, \sum_{m=1}^n \lambda_{m,\ell} \right\}$.

**Robust MDP for scheduling.** The robust MDP corresponding to scheduling can be formulated as follows. At each time, agent wishes to choose schedule $\sigma \in \mathcal{S}$ based on the system state $Q = [Q_{k,\ell}] \in \mathbb{R}_+^{n \times n}$ and earns reward equal to the negative of the sum of all queues. This reward guides the agent to minimize the average queue size.

The nature, on the other hand, determines new packet arrivals leading to changes in the system state (i.e., the queue occupancies). We restrict nature by requiring that the arrivals, or change in queue-size, belongs to set $\Lambda(x) = \{\lambda \in \mathbb{R}_+^{n \times n} : \rho(\lambda) \leq x\}$ for some fixed $x \in (0, 1)$.

The constraint $x < 1$ is an example of an *environmental constraint* imposed on the nature. This constraint is required for the packet arrival process to be *admissible*. Specifically, for $x > 1$, there is no scheduling policy that can stabilize the switch queues, because at most on packet can get serviced from any input or output port.

**Evaluation.** Given the robust MDP formulation as explained above, we apply AGZ to the corresponding transformed two-player zero-sum game, and evaluate the performance of the resulting policy. Details of our experiments can be found in Appendix G. We compare the performance of AGZ with respect to the standard SARSA policy (Rummery & Niranjan, 1994). We also compare it with Maximum Weight Matching scheduling algorithm parameterized by a parameter $\alpha > 0$ denoted as MWM-$\alpha$. MWM is a classical scheduling algorithm. Specifically, MWM-$\alpha$ chooses schedule $\sigma \in \mathcal{S}$ such that $\sigma \in \arg\max_{\tilde{\sigma} \in \mathcal{S}} \sum_{k,\ell} \tilde{\sigma}_{k,\ell} Q_{k,\ell}^\alpha$. The MWM-1 algorithm was proposed in 1990s McKeown et al. (1999) and was shown to be within factor 2 of the fundamental lower bound recently Maguluri & Srikant (2016). It has also been argued that MWM-$\alpha$ improves performance as $\alpha \to 0^+$ (Shah & Wischik, 2006).

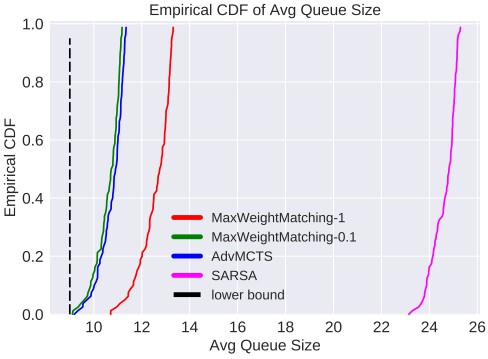

Figure 1: Empirical CDF of the avg queue size.

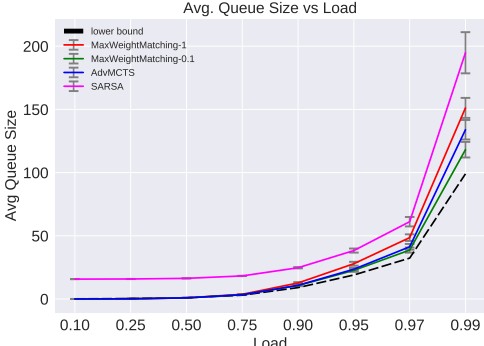

Figure 2: Avg queue size vs. different loads.

Our evaluation results are shown in Figures 1 and 2. We evaluate the AGZ scheduling policy on 80 different arrival traffic scenarios, each with load equal to 0.9. Figure 1 shows the empirical CDF of the average queue sizes on these 80 evaluations. As can be seen, the AGZ policy (called AdvMCTS) outperforms vanilla SARSA. It also outperforms MWM-1 and performs nearly as well as MWM-0.1. A similar conclusion is observed in Figure 2 where we compare performance of algorithms in terms of the average queue size at different loads varying between 0 and 1, averaged over 16 traffic arrival patterns. We note that the dashed line in both plots represents the fundamental lower bound, cf. Shah et al. (2012). This performance of AGZ is remarkable given the fact that it has taken three decades of research to arrive at similar performing algorithms in the literature.

## 6 CONCLUSION

Motivated to explain the success of AlphaGo Zero (AGZ), in this paper, we provide a formal framework to study AGZ. Specifically, we explain that AGZ attempts to learn the Nash equilibrium for two-player zero-sum games. In the process, we establish a novel bound on difference of reward earned under two different policies in terms of cross-entropy between the two policies for the setting of two-player games. By formulating generic sequential decision making through the framework of robust Markov Decision Process, we show how to view these problems as a two-player zero-sum game. This allows extending AGZ beyond games. We present a case-study for performance of AGZ for robust MDP in the context of input-queued switch scheduling. We find that AGZ performs remarkably well and its performance is nearly as good as the best-known scheduling policies.

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

# Appendices

## A  PSEUDOCODE FOR ALPHAGO ZERO

We provide the pseudocode for AlghaGo Zero in Algorithm 1, which concisely summarizes the overview in the Introduction.

---
**Algorithm 1** Training Algorithm for Each Episode
---
1:  *Sampling:* Suppose that the current episode ends in $L$ steps.
2:  **for** each step $t = 0, 1, \dots, L$ **do**
3:      run MCTS for some time (1600 simulations), with state $s_t$ as the root and the current player taking the first action; obtain the action distribution $\pi_t(\cdot|s_t)$ based on empirical visit counts.
4:      sample an action $a_t \sim \pi_t(\cdot|s_t)$ and observe a new state $s_{t+1}$ for the other player to play.
5:  **end for**
6:  *Store Data:* the episode terminates and an reward $r_L \in \{-1, 1\}$ is observed. Then, the data for each time step $t = 0, 1, \dots, T$ is stored as $(s_t, \pi_t(\cdot, |s_t), z_t)$, where $z_t = \pm r_T$ is the game winner from the perspective of the current player at step $t$.
7:  *Update Parameters:* when decide to update the parameters $\theta$, sample a mini-batch $M$ and run gradient descent:
$$\theta \leftarrow \theta - \alpha \nabla_\theta \Big[ \sum_M -\pi(\cdot|s) \log \pi_\theta(\cdot|s) + (z - v_\theta)^2 + \lambda \|\theta\|^2 \Big]$$

---

## B  RELATED WORKS

In the recent years, deep reinforcement learning has emerged as a generic yet effective learning method, achieving remarkable performance across a variety of fields ranging from game playing (Mnih et al., 2015; Jaderberg et al., 2016; Silver et al., 2017b;a), simulated physical tasks (Lillicrap et al., 2015; Schulman et al., 2015b) to chemistry (Segler et al., 2018). An important instantiation, self-play reinforcement learning, has been integrated with other techniques—including MCTS and supervised learning— to achieve superhuman performance in games without human knowledge; as exemplified by the AlphaGo Zero. The success of AlpahGo Zero has inspired some recent RL algorithms (Jiang et al., 2018; Efroni et al., 2018; Sun et al., 2018). However, exactly capturing AlphaGo Zero and understanding the success of it is very limited beyond natural intuitions. To our best knowledge, this is the first attempt towards a principled and theoretical understanding for its underlying ingredients, under minimal assumptions.

The ideal variant of AlphaGo Zero falls into the general category of policy-based methods for reinforcement learning (Williams, 1992; Sutton et al., 2000; Marbach & Tsitsiklis, 2001; Kakade, 2002; Greensmith et al., 2004; Schulman et al., 2015a), a line of research that receives considerable interest due to its advantage of directly parameterizing the policy. Most of the prior algorithms were obtained via *directly* optimizing the expected return $R(\pi_\theta)$ of a parameterized policy. For example, the classical REINFORCE algorithm (Williams, 1992) attempts to maximize the return by gradient ascent with unbiased estimator of the gradient $\nabla R(\pi_\theta)$. The recent TRPO algorithm (Schulman et al., 2015a) attempts to maximize a lower bound of $R(\pi)$. In contrast, the algorithm of AlphaGo Zero that we study approaches the problem *indirectly*: it minimizes the KL divergence (or cross entropy) between current policy $\pi$ and optimal policy $\pi^*$; in this work, we argue that this step corresponds to minimizes a *valid* upper bound on the difference between $R(\pi)$ and the optimal $R(\pi^*)$.

In the practical version of the algorithm, idealized oracle is replaced with MCTS, which, by itself, is a well studied and widely applicable technique (Chang et al., 2005; Coulom, 2006; Kocsis & Szepesvári, 2006; Kocsis et al., 2006; Browne et al., 2012). In particular, MCTS has been shown to asymptotically converge to the optimal actions in both discounted MDPs (Chang et al., 2005; Kocsis & Szepesvári, 2006) and two-player zero-sum games (Kocsis et al., 2006). However, the interplay between policy-based learning methods and MCTS, investigated in this paper, is new.

We note that our results are applicable in the context of the two-player stochastic games which have been extensively studied in the literature, see (Littman, 1994; Patek, 1997; Hansen et al., 2013; Perolat et al., 2015).

## C  PRELIMINARY FACTS

The following inequalities are used for developing our technical results:

**Jensen's Inequality:** Let $X$ be a random variable and $\phi$ be a convex function, then $\phi(\mathbb{E}[X]) \leq \mathbb{E}[\phi(X)]$.

**Pinsker's Inequality:** Let $\mu$ and $\nu$ be two probability distributions, then the total variation distance $TV(\mu, \nu)$ and the KL divergences $D_{\mathrm{KL}}(\mu\|\mu)$ satisfies

$$TV(\mu, \nu) \leq \sqrt{\frac{1}{2} D_{\mathrm{KL}}(\mu, \nu)}$$

Note that if $\mu$ and $\nu$ are discrete distributions, then $TV(\mu, \nu) = \frac{1}{2} \sum_{\omega \in \Omega} |\mu(\omega) - \nu(\omega)|$.

## D  PROOF OF THEOREM 2

We define the state-action value or $Q$ function associated with policy $\pi = (\pi_1, \pi_2)$ as follows: for each $s \in \mathcal{S}, a \in \mathcal{A}$,

$$Q_{\pi_1, \pi_2}(s, a) = \mathbb{E}_{s_{t+1}, a_{t+1}, \ldots} \left[ \sum_{k=0}^{\infty} \gamma^k r(s_{t+k}, a_{t+k}) | s_t = s, a_t = a \right].$$

Define the corresponding *advantage function* $A_{\pi_1, \pi_2}$: for each $s \in \mathcal{S}, a \in \mathcal{A}$,

$$A_{\pi_1, \pi_2}(s, a) = Q_{\pi_1, \pi_2}(s, a) - V_{\pi_1, \pi_2}(s) \tag{9}$$

Recall that the reward function $r(\cdot, \cdot)$ is assumed to be uniformly bounded by some constant $C_R$. An immediate consequence is that for any policy $\pi = (\pi_1, \pi_2)$, $s \in \mathcal{S}, a \in \mathcal{A}$,

$$|A_{\pi_1, \pi_2}(s, a)| = |Q_{\pi_1, \pi_2}(s, a) - V_{\pi_1, \pi_2}(s)|$$
$$\leq \frac{2}{1 - \gamma} \max_{s' \in \mathcal{S}, a' \in \mathcal{A}} |r(s', a')| = \frac{2C_R}{1 - \gamma}.$$

That is, $|A_{\pi_1, \pi_2}(s, a)| \leq C_A = \frac{2C_R}{1-\gamma}$ for each $s \in \mathcal{S}$ and $a \in \mathcal{A}$.

For two policies $\tilde{\pi} = (\tilde{\pi}_1, \tilde{\pi}_2)$ and $\pi = (\pi_1, \pi_2)$, we can express the difference in policy performance $R(\tilde{\pi}) - R(\pi)$, as a sum of per-timestep advantages over $\pi$, in the same fashion as Kakade and Langford's result (Kakade & Langford, 2002) for MDP. This is done in the following lemma, whose proof is given in Appendix D.1.

**Lemma 3.** *Given any two policies $\tilde{\pi} = (\tilde{\pi}_1, \tilde{\pi}_2)$ and $\pi = (\pi_1, \pi_2)$, we have*

$$R(\tilde{\pi}) = R(\pi) + \mathbb{E}_{\tau \sim \tilde{\pi}} \left[ \sum_{t=0}^{\infty} \gamma^t A_\pi(s_t, a_t) \right], \tag{10}$$

*where the expectation is taken over trajectories $\tau := (s_0, a_0, s_1, a_1 \ldots)$, and the notation $\mathbb{E}_{\tau \sim \tilde{\pi}}[\cdot]$ indicates that actions are sampled from policy $\tilde{\pi} = (\tilde{\pi}_1, \tilde{\pi}_2)$ to generate $\tau$.*

Using definition of unnormalized visit frequency (cf. (4)) we can rewrite (10) as follows:

$$R(\tilde{\pi}) = R(\pi) + \sum_{t=0}^{\infty} \sum_{s} P(s_t = s|\tilde{\pi}) \left[ \sum_{a} \tilde{\pi}_{I(s)}(a|s) \gamma^t A_\pi(s, a) \right]$$
$$= R(\pi) + \sum_{s} \sum_{t=0}^{\infty} \gamma^t P(s_t = s|\tilde{\pi}) \left[ \sum_{a} \tilde{\pi}_{I(s)}(a|s) A_\pi(s, a) \right]$$
$$= R(\pi) + \sum_{s} \rho_{\tilde{\pi}}(s) \sum_{a} \tilde{\pi}_{I(s)}(a|s) A_\pi(s, a). \tag{11}$$

We are now ready to prove Theorem 2. We shall focus on proving the inequality (5) for the optimal policy $\pi^*$. The proof for a general policy $\tilde{\pi}$ can be done by replacing $\pi^*$ with $\tilde{\pi}$. We shall use $\hat{\pi}$ as a "reference" policy in the remainder of the proof.

By taking $\tilde{\pi} = \pi^*$ and $\pi = \hat{\pi}$ in (11), we obtain

$$R(\pi^*) = R(\hat{\pi}) + \sum_s \rho_{\pi^*}(s) \sum_a \pi^*_{I(s)}(a|s) A_{\hat{\pi}}(s, a). \tag{12}$$

By taking $\tilde{\pi} = \pi$ and $\pi = \hat{\pi}$ in (11), we obtain

$$R(\pi) = R(\hat{\pi}) + \sum_s \rho_\pi(s) \sum_a \pi_{I(s)}(a|s) A_{\hat{\pi}}(s, a). \tag{13}$$

Therefore, we have

$$\left| R(\pi) - R(\pi^*) \right|$$

$$= \left| \sum_s \sum_a \left( \rho_\pi(s) \pi_{I(s)}(a|s) - \rho_{\pi^*}(s) \pi^*_{I(s)}(a|s) \right) A_{\hat{\pi}}(s, a) \right|$$

$$\leq C_A \sum_s \sum_a \left| \rho_\pi(s) \pi_{I(s)}(a|s) - \rho_{\pi^*}(s) \pi^*_{I(s)}(a|s) \right|$$

$$= C_A \sum_s \sum_a \left| \rho_\pi(s) \pi_{I(s)}(a|s) - \rho_{\pi^*}(s) \pi_{I(s)}(a|s) + \rho_{\pi^*}(s) \pi_{I(s)}(a|s) - \rho_{\pi^*}(s) \pi^*_{I(s)}(a|s) \right|$$

$$\leq C_A \sum_s \sum_a \pi_{I(s)}(a|s) \left| \rho_\pi(s) - \rho_{\pi^*}(s) \right| + C_A \sum_s \sum_a \rho_{\pi^*}(s) \left| \pi_{I(s)}(a|s) - \pi^*_{I(s)}(a|s) \right|$$

$$= C_A \sum_s \left| \rho_\pi(s) - \rho_{\pi^*}(s) \right| + C_A \sum_s \sum_a \rho_{\pi^*}(s) \left| \pi_{I(s)}(a|s) - \pi^*_{I(s)}(a|s) \right| \tag{14}$$

*First term in R.H.S. of* (14)*:* We can rewrite $\rho_\pi$ as follows

$$\rho_\pi(s) = P(S_0 = s) + \sum_{t=1}^\infty \gamma^t P(S_t = s|\pi)$$

$$= P(S_0 = s) + \sum_{t=1}^\infty \gamma^t \left[ \sum_{s'} P(S_t = s|\pi, S_{t-1} = s') P(S_{t-1} = s'|\pi) \right]$$

$$= P(S_0 = s) + \sum_{t=1}^\infty \gamma^t \left[ \sum_{s'} \left( \sum_a \pi_{I(s')}(a|s') P(s|s', a) \right) P(S_{t-1} = s'|\pi) \right]$$

$$= P(S_0 = s) + \sum_{s'} \sum_a \pi_{I(s')}(a|s') P(s|s', a) \left[ \sum_{t=1}^\infty \gamma^t P(S_{t-1} = s'|\pi) \right]$$

$$= P(S_0 = s) + \gamma \sum_{s'} \sum_a \pi_{I(s')}(a|s') P(s|s', a) \rho_\pi(s').$$

As a consequence, we have

$$\left| \rho_\pi(s) - \rho_{\pi^*}(s) \right| = \gamma \left| \sum_{s'} \sum_a \pi_{I(s')}(a|s') P(s|s', a) \rho_\pi(s') - \sum_{s'} \sum_a \pi^*_{I(s')}(a|s') P(s|s', a) \rho_{\pi^*}(s') \right|$$

$$\leq \gamma \left| \sum_{s'} \sum_a \pi_{I(s')}(a|s') P(s|s', a) \rho_\pi(s') - \sum_{s'} \sum_a \pi_{I(s')}(a|s') P(s|s', a) \rho_{\pi^*}(s') \right|$$

$$+ \gamma \left| \sum_{s'} \sum_a \pi_{I(s')}(a|s') P(s|s', a) \rho_{\pi^*}(s') - \sum_{s'} \sum_a \pi^*_{I(s')}(a|s') P(s|s', a) \rho_{\pi^*}(s') \right|$$

$$\leq \gamma \sum_{s'} \sum_a \pi_{I(s')}(a|s') P(s|s', a) \left| \rho_\pi(s') - \rho_{\pi^*}(s') \right|$$

$$+ \gamma \sum_{s'} \sum_a P(s|s', a) \rho_{\pi^*}(s') \left| \pi_{I(s')}(a|s') - \pi^*_{I(s')}(a|s') \right|.$$

Summation over $s \in S$ on both sides yields

$$\sum_s |\rho_\pi(s) - \rho_{\pi^*}(s)| \leq \gamma \sum_s \sum_{s'} \sum_a \pi_{I(s')}(a|s')P(s|s',a)|\rho_\pi(s') - \rho_{\pi^*}(s')|$$
$$+ \gamma \sum_s \sum_{s'} \sum_a P(s|s',a)\rho_{\pi^*}(s')|\pi_{I(s')}(a|s') - \pi^*_{I(s')}(a|s')|$$
$$= \gamma \sum_{s'} |\rho_\pi(s') - \rho_{\pi^*}(s')|\left[\sum_a \pi_{I(s')}(a|s')\left(\sum_s P(s|s',a)\right)\right]$$
$$+ \gamma \sum_{s'} \sum_a \rho_{\pi^*}(s')|\pi_{I(s')}(a|s') - \pi^*_{I(s')}(a|s')|\left(\sum_s P(s|s',a)\right)$$
$$= \gamma \sum_{s'} |\rho_\pi(s') - \rho_{\pi^*}(s')| + \gamma \sum_{s'} \sum_a \rho_{\pi^*}(s')|\pi_{I(s')}(a|s') - \pi^*_{I(s')}(a|s')|.$$

To conclude, we have

$$\sum_s |\rho_\pi(s) - \rho_{\pi^*}(s)| \leq \frac{\gamma}{1-\gamma} \sum_{s'} \sum_a \rho_{\pi^*}(s')|\pi_{I(s')}(a|s') - \pi^*_{I(s')}(a|s')|. \tag{15}$$

*Concluding proof of Theorem 2:* Substituting (15) into (14) yields

$$|R(\pi) - R(\pi^*)| \leq \frac{C_A}{1-\gamma} \sum_s \sum_a \rho_{\pi^*}(s)|\pi_{I(s)}(a|s) - \pi^*_{I(s)}(a|s)|$$
$$= \frac{C_A}{(1-\gamma)^2} \sum_s (1-\gamma)\rho_{\pi^*}(s) \sum_a |\pi_{I(s)}(a|s) - \pi^*_{I(s)}(a|s)|$$
$$= \frac{2C_A}{(1-\gamma)^2} \mathbb{E}_{s\sim\bar\rho_{\pi^*}}\left[TV\left(\pi^*_{I(s)}(\cdot|s)\|\pi_{I(s)}(\cdot|s)\right)\right]$$
$$\overset{(a)}{\leq} \frac{\sqrt{2}C_A}{(1-\gamma)^2} \mathbb{E}_{s\sim\bar\rho_{\pi^*}}\left[\sqrt{D_{\mathrm{KL}}\left(\pi^*_{I(s)}(\cdot|s)\|\pi_{I(s)}(\cdot|s)\right)}\right]$$
$$\overset{(b)}{\leq} \frac{\sqrt{2}C_A}{(1-\gamma)^2} \sqrt{\mathbb{E}_{s\sim\bar\rho_{\pi^*}}\left[D_{\mathrm{KL}}\left(\pi^*_{I(s)}(\cdot|s)\|\pi_{I(s)}(\cdot|s)\right)\right]},$$

where (a) follows from Pinker's inequality and (b) follows from Jensen's inequality. This concludes the proof of Theorem 2.

### D.1 PROOF OF LEMMA 3

By the definition of the advantage function (cf. 9), for each policy $\pi = (\pi_1, \pi_2)$,

$$A_\pi(s,a) = Q_\pi(s,a) - V_\pi(s)$$
$$= \mathbb{E}_{s'\sim P(s,a)}\left[r(s,a) + \gamma V_\pi(s') - V_\pi(s)\right].$$

Therefore,

$$\mathbb{E}_{\tau\sim\tilde\pi}\left[\sum_{t=0}^\infty \gamma^t A_\pi(s_t, a_t)\right] = \mathbb{E}_{\tau\sim\tilde\pi}\left[\sum_{t=0}^\infty \gamma^t\left(r(s_t,a_t) + \gamma V_\pi(s_{t+1}) - V_\pi(s_t)\right)\right]$$
$$= \mathbb{E}_{\tau\sim\tilde\pi}\left[\sum_{t=0}^\infty \gamma^t r(s_t,a_t) - V_\pi(s_0)\right] \tag{16}$$
$$= \mathbb{E}_{\tau\sim\tilde\pi}\left[\sum_{t=0}^\infty \gamma^t r(s_t,a_t)\right] - \mathbb{E}_{s_0}\left[V_\pi(s_0)\right]$$
$$= R(\tilde\pi) - R(\pi).$$

In above, (16) is the result of telescoping summation.

# E   STOCHASTIC GAMES

A Stochastic Game (SG) (also called Markov Games) is a generalization of an MDP to a $n$-player setting. SGs are widely used to model sequential decision making in multi-agent systems. Each player controls the SG through a set of actions. In the case of a *turn-based* game—we use it to model the game of Go—each state is controlled by a single player. In a general SG, at each step of the game, all players *simultaneously* choose an action. The reward each player gets after one step depends on the state and the joint action of all players. Furthermore, the transition kernel of the SG is controlled jointly by all the players.

Consider a zero-sum two-player SG. We use the same notation for the two players as the turn-based game, i.e., P1 and P2 for player 1 and player 2, respectively. This game can be described by a tuple $(\mathcal{S}, \mathcal{A}_1, \mathcal{A}_2, P, r, \rho_0, \gamma)$ where $\mathcal{A}_1(s)$ and $\mathcal{A}_2(s)$ are the set of actions player 1 and player 2 can take in state $s$ respectively; $P : \mathcal{S} \times \mathcal{A}_1 \times \mathcal{A}_2 \times \mathcal{S} \to \mathbb{R}$ is the state transition probability; $r : \mathcal{S} \times \mathcal{A}_1 \times \mathcal{A}_2 \to \mathbb{R}$ is the reward for both players; $\rho_0$ is the distribution of the initial state $s_0$ and $\gamma$ is the discounted factor. Let $\pi_i : \mathcal{S} \times \mathcal{A}_i \to [0, 1]$ denote the randomized policy of player $i$, with $\pi_i(a|s)$ denoting the probability of taking action $a$ given state $s$. This formulation has been used to model adversarial reinforcement learning Pinto et al. (2017).

At each timestep $t$, both players observe the state $s_t$ and take actions $a_t^1 \sim \pi_1(\cdot|s_t)$ and $a_t^2 \sim \pi_2(\cdot|s_t)$. In the zero-sum game, without loss of generosity we assume that P1 gets a reward $r_t^1 = r_t = r(s_t, a_t^1, a_t^2)$ while P2 gets a reward $r_t^2 = -r_t$. If P1 and P2 respectively use the stationary policies $\pi_1$ and $\pi_2$, the cumulative reward for P1 is given by

$$R^1(\pi_1, \pi_2) = R(\pi_1, \pi_2) = \mathbb{E}_{\substack{s_0 \sim \rho_0 \\ a_t^1 \sim \pi_1(\cdot|s_t) \\ a_t^2 \sim \pi_2(\cdot|s_t) \\ s_{t+1} \sim P(\cdot|s_t, a_t^1, a_t^2)}} \left[ \sum_{t=0}^{\infty} \gamma^t r(s_t, a_t^1, a_t^2) \right].$$

Each player tries to maximize his discounted cumulative reward. In this formulation, P1 aims to maximize $R^1(\pi_1, \pi_2)$ while the goal of P2 is to minimize it. In a zero-sum two player SG, it has been shown that the pairs of optimal policies conincide with the Nash equilibrium of this game. That is, the value attained at the minimax equilibrium is the optimal value:

$$R^* = \min_{\pi_1} \max_{\pi_2} R^1(\pi_1, \pi_2) = \max_{\pi_2} \min_{\pi_1} R^1(\pi_1, \pi_2).$$

Let $\pi^* = (\pi_1^*, \pi_2^*)$ denote the pair of optimal policies that attain the optimal value.

The result for two-player stochastic game setting is a similar upper bound as in the turn-based game case.

**Theorem 4.** *In a two-player zero-sum SG, for a policy pair $\pi = (\pi_1, \pi_2)$,*

$$|R(\pi_1, \pi_2) - R(\pi_1^*, \pi_2^*)| \leq C \sqrt{\mathbb{E}_{s \sim \bar{\rho}_{\pi^*}} \left[ D_{\mathrm{KL}}\big(\pi_1^*(\cdot|s) \| \pi_1(\cdot|s)\big) + D_{\mathrm{KL}}\big(\pi_2^*(\cdot|s) \| \pi_2(\cdot|s)\big) \right]}. \quad (17)$$

*where $C$ is a constant depending only on $\gamma$ and $C_R$.*

The proof of Theorem 4 follows the same line of argument as that of Theorem 2. We use the following standard definitions of the state-action value function $Q_{\pi_1, \pi_2}$, the value function $V_{\pi_1, \pi_2}$ and the advantage function $A_{\pi_1, \pi_2}$:

$$Q_{\pi_1, \pi_2}(s, a, a') = \mathbb{E}_{s_{t+1}, a_{t+1}, a'_{t+1} \cdots} \left[ \sum_{k=0}^{\infty} \gamma^k r(s_{t+k}, a_{t+k}, a'_{t+k}) | s_t = s, a_t = a, a'_t = a' \right],$$

$$V_{\pi_1, \pi_2}(s) = \mathbb{E}_{a_t, s_{t+1}, a_{t+1}, \cdots} \left[ \sum_{k=0}^{\infty} \gamma^k r(s_{t+k}, a_{t+k}, a'_{t+k}) | s_t = s \right],$$

$$A_{\pi_1, \pi_2}(s, a, a') = Q_{\pi_1, \pi_2}(s, a, a') - V_{\pi_1, \pi_2}(s).$$

Like in the setting of turn-based games, the advantage function is uniformly bounded. Specifically, for any policy $\pi = (\pi_1, \pi_2)$, $s \in \mathcal{S}, a \in \mathcal{A}, a' \in \mathcal{A}$,

$$|A_{\pi_1, \pi_2}(s, a, a')| = |Q_{\pi_1, \pi_2}(s, a, a') - V_{\pi_1, \pi_2}(s)| \leq \frac{2}{1 - \gamma} \max_{s \in \mathcal{S}, a \in \mathcal{A}, a' \in \mathcal{A}} |r(s, a, a')| = \frac{2C_R}{1 - \gamma}.$$

Similar to the turn-based game setting, we can express the difference in policy performance $R(\tilde{\pi}) - R(\pi)$ as a sum of per-timestep advantages over $\pi$, as stated in the following lemma. The proof is similar to that of Lemma 3 and is omitted here.

**Lemma 5.** *Given any two policies $\tilde{\pi} = (\tilde{\pi}_1, \tilde{\pi}_2)$ and $\pi = (\pi_1, \pi_2)$, we have*

$$R(\tilde{\pi}) = R(\pi) + \mathbb{E}_{\tau \sim \tilde{\pi}}\left[\sum_{t=0}^{\infty} \gamma^t A_\pi(s_t, a_t, a_t')\right] \tag{18}$$

*where the expectation is taken over trajectories $\tau := (s_0, a_0, s_1, a_1 \ldots)$, and the notation $\mathbb{E}_{\tau \sim \tilde{\pi}}[\cdot]$ indicates that actions are sampled from policy $\tilde{\pi}$ to generate $\tau$.*

Let $\rho_\pi$ be the unnormalized discounted visit frequencies, defined the same as (4):

$$\rho_\pi(s) = P(s_0 = s) + \gamma P(s_1 = s) + \gamma^2 P(s_2 = s) + \ldots,$$

Consistently, let $\bar{\rho}_\pi$ be the corresponding normalized probability distribution. Then, analogous to the turn-based game setting, we can rewrite (18) as follows:

$$\begin{aligned}
R(\tilde{\pi}) &= R(\pi) + \sum_{t=0}^{\infty}\sum_s P(s_t = s|\tilde{\pi})\left[\sum_a\sum_{a'}\tilde{\pi}_1(a|s)\tilde{\pi}_2(a'|s)\gamma^t A_\pi(s, a, a')\right] \\
&= R(\pi) + \sum_s\sum_{t=0}^{\infty}\gamma^t P(s_t = s|\tilde{\pi})\left[\sum_a\sum_{a'}\tilde{\pi}_1(a|s)\tilde{\pi}_2(a'|s) A_\pi(s, a, a')\right] \\
&= R(\pi) + \sum_s \rho_{\tilde{\pi}}(s)\sum_a\sum_{a'}\tilde{\pi}_1(a|s)\tilde{\pi}_2(a'|s) A_\pi(s, a, a').
\end{aligned} \tag{19}$$

Now (19) implies equalities similar to (12) and (13). Using a policy $\hat{\pi}$ as the reference policy, we can bound $R(\pi) - R(\pi^*)$ as follows:

$$\begin{aligned}
&\left|R(\pi) - R(\pi^*)\right| \\
=&\left|\left(R(\pi) - R(\hat{\pi})\right) - \left(R(\pi^*) - R(\hat{\pi})\right)\right| \\
=&\left|\sum_s\sum_a\sum_{a'}\left(\rho_\pi(s)\pi_1(a|s)\pi_2(a'|s) - \rho_{\pi^*}(s)\pi_1^*(a|s)\pi_2^*(a'|s)\right) A_{\hat{\pi}}(s, a, a')\right| \\
\leq& C_A\sum_s\sum_a\sum_{a'}\left|\rho_\pi(s)\pi_1(a|s)\pi_2(a'|s) - \rho_\pi^*(s)\pi_1(a|s)\pi_2(a'|s)\right. \\
&\left. + \rho_\pi^*(s)\pi_1(a|s)\pi_2(a'|s) - \rho_{\pi^*}(s)\pi_1^*(a|s)\pi_2^*(a'|s)\right| \\
\leq& C_A\sum_s\left|\rho_\pi(s) - \rho_{\pi^*}(s)\right| + C_A\sum_s\sum_a\sum_{a'}\rho_{\pi^*}(s)\left|\pi_1(a|s)\pi_2(a'|s) - \pi_1^*(a|s)\pi_2^*(a'|s)\right| \tag{20}
\end{aligned}$$

Following identical steps of arguments utilized to bound R.H.S. of (14) in the proof of Theorem 2, we can obtain the bound on the *first term in the R.H.S.* of (20) as follows:

$$\sum_s\left|\rho_\pi(s) - \rho_{\pi^*}(s)\right| \leq \frac{\gamma}{1-\gamma}\sum_{s'}\sum_a\sum_{a'}\rho_{\pi^*}(s')\left|\pi_1(a|s')\pi_2(a'|s') - \pi_1^*(a|s')\pi_2^*(a'|s')\right|. \tag{21}$$

For each $s \in \mathcal{S}, a \in \mathcal{A}_1, a' \in \mathcal{A}_2$, we use the notation

$$\pi(a, a'|s) = \pi_1(a|s)\pi_2(a'|s)$$

to represent the joint action distribution of the two players under the policy $(\pi_1, \pi_2)$. Substituting (21) into (20) yields

$$
\begin{aligned}
\left|R(\pi) - R(\pi^*)\right| &\leq \frac{C_A}{1-\gamma} \sum_s \sum_a \sum_{a'} \rho_{\pi^*}(s) \left|\pi_1(a|s)\pi_2(a'|s) - \pi_1^*(a|s)\pi_2^*(a'|s)\right| \\
&= \frac{C_A}{(1-\gamma)^2} \sum_s (1-\gamma)\rho_{\pi^*}(s) \sum_a \sum_{a'} \left|\pi_1(a|s)\pi_2(a'|s) - \pi_1^*(a|s)\pi_2^*(a'|s)\right| \\
&= \frac{2C_A}{(1-\gamma)^2} \mathbb{E}_{s \sim \bar{\rho}_{\pi^*}} \left[TV\left(\pi^*(\cdot,\cdot|s) \| \pi(\cdot,\cdot|s)\right)\right] \\
&\overset{(a)}{\leq} \frac{\sqrt{2}C_A}{(1-\gamma)^2} \mathbb{E}_{s \sim \bar{\rho}_{\pi^*}} \left[\sqrt{D_{\mathrm{KL}}\left(\pi^*(\cdot,\cdot|s) \| \pi(\cdot,\cdot|s)\right)}\right] \\
&\overset{(b)}{\leq} \frac{\sqrt{2}C_A}{(1-\gamma)^2} \sqrt{\mathbb{E}_{s \sim \bar{\rho}_{\pi^*}} \left[D_{\mathrm{KL}}\left(\pi^*(\cdot,\cdot|s) \| \pi(\cdot,\cdot|s)\right)\right]} \\
&= \frac{\sqrt{2}C_A}{(1-\gamma)^2} \sqrt{\mathbb{E}_{s \sim \bar{\rho}_{\pi^*}} \left[D_{\mathrm{KL}}\left(\pi_1^*(\cdot|s) \| \pi_1(\cdot|s)\right) + D_{\mathrm{KL}}\left(\pi_2^*(\cdot|s) \| \pi_2(\cdot|s)\right)\right]},
\end{aligned}
$$

where (a) follows from Pinker's inequality and (b) follows from Jensen's inequality. This concludes the proof of Theorem 4.

## F   ADDITIONAL DISCUSSIONS ON THEOREM 2 AND THE IDEAL ALGORITHM

Theorem 2 translates the distance between a trained policy $\pi$ and the optimal one to the distance of their expected rewards. It reinforces the intuition that if the expected KL divergence for the two policies is small, then the difference of respected rewards should also be small. Crucially, note that the expectation is taken with respect to $\bar{\rho}_{\pi^*}$. This fact allows one to use Monte Carlo methods to run stochastic gradient descent (SGD) for policy optimization (i.e., the upper bound), and hence, obtain a higher expected reward. The following pseudocode summarizes the ideal learning algorithm discussed in Section 4.1, which is suggested by our Theorem 2.

---
**Algorithm 2** Supervised Policy Optimization (theoretical)
---
1:  Optimizing a parameterized policy $(\pi_{\theta_1}, \pi_{\theta_2})$
2:  **for** each step $t = 0, 1, 2, \ldots$ **do**
3:      query an oracle $\mathcal{O}$ at state $s_t$, which returns the optimal policy at state $s_t$, i.e., $\pi_{I(s)}^*(\cdot|s_t)$
4:      run SGD to optimize the KL divergence between $\pi_{I(s)}^*(\cdot|s_t)$ and $\pi_{\theta_{I(s)}}(\cdot|s)$

$$
\theta_{I(s_t)} \leftarrow \theta_{I(s_t)} - \alpha_t \nabla_\theta D_{\mathrm{KL}}\left(\pi_{I(s_t)}^*(\cdot|s_t) \| \pi_{\theta_{I(s_t)}}(\cdot|s_t)\right) \tag{22}
$$

5:      sample an action $a_t \sim \pi_{I(s_t)}^*(\cdot|s_t)$ and observe a new state $s_{t+1}$
6:  **end for**
---

To better appreciate the power of Algorithm 2 and Theorem 2, let us consider two closely related metrics that also upper bound $|R(\pi) - R(\pi^*)|$. Interchanging the roles of $\pi^*$ and $\pi$ in (5) gives the bound

$$
\left|R(\pi) - R(\pi^*)\right| \leq C \sqrt{\mathbb{E}_{s \sim \bar{\rho}_\pi} \left[D_{\mathrm{KL}}\left(\pi_{I(s)}(\cdot|s) \| \pi_{I(s)}^*(\cdot|s)\right)\right]},
$$

which suggests optimizing the metric $\mathbb{E}_{s \sim \bar{\rho}_\pi} \left[D_{\mathrm{KL}}\left(\pi_{I(s)}(\cdot|s) \| \pi_{I(s)}^*(\cdot|s)\right)\right]$. Doing so is, however, of little use: since optimal policy $\pi_{I(s)}^*(\cdot|s)$ is supported only on the optimal actions, the divergence $D_{\mathrm{KL}}\left(\pi_{I(s)}(\cdot|s) \| \pi_{I(s)}^*(\cdot|s)\right)$ is always $+\infty$ unless $\pi_{I(s)}(\cdot|s)$ is also optimal.

Another possibility is to use the bound

$$
\left|R(\pi) - R(\pi^*)\right| \leq C \sqrt{\mathbb{E}_{s \sim \bar{\rho}_\pi} \left[D_{\mathrm{KL}}\left(\pi_{I(s)}^*(\cdot|s) \| \pi_{I(s)}(\cdot|s)\right)\right]}, \tag{23}
$$

which follows from the same arguments as for Theorem 2. This bound differs from (5) in that the expectation is taken with respect to $\bar{\rho}_\pi$ rather than $\bar{\rho}_{\pi^*}$. Therefore, to minimize the R.H.S. of (23), one

would run SGD to update $\theta$ by sampling from the current policy $\pi_\theta$ rather than from the optimal $\pi^*$. As the minimization of $D_{\mathrm{KL}}\big(\pi^*_{I(s)}(\cdot|s)\|\pi_{\theta_{I(s)}}(\cdot|s)\big)$ can be done only approximately, it is important to focus on those states $s$ frequently visited by the optimal policy—these are important states that tend to have higher values and hence have a larger impact on the return of a policy. Therefore, optimizing the R.H.S. of (5) as done in Algorithm 2, rather than that of (23), is more likely to require less samples and produce a solution with better generalization performance.

Additionally, we note that the bound (5) is no longer valid if $\bar\rho_{\pi^*}$ is replaced by an arbitrary distribution $\bar\rho$. As an example, pick a particular state $s_0$, and assume that $\bar\rho(s_0) = 1$. Suppose that $\pi$ equals $\pi^*$ at $s_0$, but otherwise takes extremely bad actions. One sees that in this case $\mathbb{E}_{s\sim\bar\rho}\big[D_{\mathrm{KL}}\big(\pi^*_{I(s)}(\cdot|s)\|\pi_{I(s)}(\cdot|s)\big)\big] = 0$, but $R(\pi)$ may be very far from $R(\pi^*)$. This observation further demonstrates the delicacy of the bound in Theorem 2 and the importance of sampling from $\bar\rho_{\pi^*}$.

Before closing, let us remark on a subtle difference between the upper bound of Theorem 2 and what is actually optimized by the ideal algorithm, Algorithm 2. Note that based on Theorem 2, we are interested in minimizing $\mathbb{E}_{s\sim\bar\rho_{\pi^*}}\Big[D_{\mathrm{KL}}\big(\pi^*_{I(s)}(\cdot|s)\|\pi_{I(s)}(\cdot|s)\big)\Big]$. That is, if one was to use SGD, the state should be sampled from the *discounted* visit frequencies $\bar\rho_{\pi^*}$. In contrast, Algorithm 2 samples actions from $\mu^*$; that is, states are sampled under the distribution induced by $\pi^*$ without discounting. However, this difference can be essentially ignored as we argue now. Let $\mu^*$ be the stationary distribution under policy $\pi^*$, i.e., $\mu^*(s) = \lim_{t\to\infty} P(s_t = s)$. Notice that as $\gamma \to 1$, by the definition of $\rho_{\pi^*}(s)$ in (4), $\bar\rho_{\pi^*}(s) \triangleq (1-\gamma)\rho_{\pi^*}(s) \to \mu^*(s)$. In practice, large $\gamma$ ($\gamma \approx 1$) is used. Therefore, the difference between the two is very minimal, which justifies that the procedure used in Algorithm 2 is a valid approximation.

Finally, as discussed in Section 4.2, in practice, the optimal oracle is replaced with an approximate version, with MCTS being one particular example. In addition, We also note that in developing such an algorithm, one could have parametrized the two policies separately in the form $\pi = (\pi_{\theta_1}, \pi_{\theta_2})$, as presented in the ideal algorithm above, by using for example one neural network for each player's policy. Depending on current player $I(s)$ at state $s$, one of the networks is chosen and the output is $\pi_{I(s)}(\cdot|s)$. However, it is also valid if we add $I(s)$ into the input, that is, one can use only one neural network: the input to it is $(s, I(s))$ and the output is interpreted as $\pi_\theta(\cdot|s, I(s))$, the policy for player $I(s)$ at state $s$. The theory still holds provided the network is expressive enough. The two polices $\pi_1$ and $\pi_2$ now share the same parameters $\theta$. Such a joint parametrization corresponds to the self-play setting in AlphaGo Zero, where a single network is used and the input to it is both the current board and the player to play.

# G    SWITCH SCHEDULING

## G.1    BASIC SETUP

Switch scheduling is a classical problem where at each time slot, a controller selects a matching between the $n$ input ports and the $n$ output ports to transmit packets. Figure 3 illustrates an example of an input-queued switch with three input ports and three output ports. Packets arriving at input $k$ destined for output $\ell$ are stored at input port $k$, in queue $Q_{k,\ell}$, thus there are $N = 9$ queues in total. We consider the case where the new data packets are arriving at queue $(k, l)$ at rate $\lambda_{k,l}$ for $1 \le k, l \le n$, according to Bernoulli distributions. That is, the number of packets arriving at queue $Q_{ij}$ is a Bernoulli random variable with mean $\lambda_{ij}$. We call the matrix $\lambda$ the traffic matrix, and let us denote by $\Lambda(x)$ the set of traffic matrices with effective load less than or equal to $x$. Overall, the queue length for each queue evolves according to:

$$Q_{ij}(t + 1) = \max\big\{Q_{ij}(t) - \sigma_{ij}(t), 0\big\} + \mathrm{Ber}(\lambda_{ij}(t)), \tag{24}$$

where $\sigma$ is the matching selected by the controller.

To train a robust controller, we model the nature, i.e., the arrival processes that determine the state transitions, as the adversary (the second player), whose action determines the packets' arrival rates. With this two-player setup, we now introduce the relevant quantities used during training: (a) Naturally, the state $S$ of the switch is the current queue lengths, i.e., $S = \{Q_{ij}\}_{1\le i,j\le N}$. (b) The action $a$ of the controller is a matching $\sigma$ between the input ports and the output ports. For any state

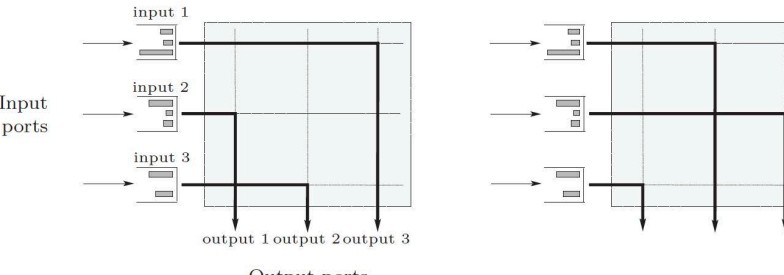

Figure 3: An input-queued switch, and two example matchings of inputs to outputs. Packets arriving at input $k$ destined for output $\ell$ are stored at input port $k$, in queue $Q_{k,\ell}$, thus with three input ports and three output ports, there are $N = 9$ queues in total.

$S$, the set of feasible actions for the controller, $\mathcal{A}(S)$, is exactly the set of possible matchings. (c) The action $\tilde{a}$ of the adversary is a traffic matrix $\lambda$, and the set of actions of the adversary is the set of traffic matrices with load less than or equal to $x$, i.e., $\mathcal{A}_d(S) = \Lambda(x)$. (d) At state $S$, the reward $r^1(S, A)$ for the controller is the negative total queue lengths after an action $A$ (played either by the controller or the nature) is applied, i.e., $r(S, A) \triangleq -\sum_{i,j}(q_{ij}$ of state $S')$, where $S'$ is the state after an action is applied. Naturally, this reward function encourages the controller to keep queues small and hence, smaller packet delay. On the other hand, the adversary's goal is to compete with the controller by increasing the average queue size. The reward $r^2(S, A)$ for the adversary is always equal to $-r^1(S, A)$.

As a proof of concept, we consider a 3-port instance in our experiment. This means that we have 9 queues in total. We also restrict the maximum queue size to be 100. At first glance, $n = 3$ may seem small. However, the system actually has a huge state space with $10^{18}$ configurations. Overall the scheduling problem is hard, even for seemingly small instances. When $n = 3$, there are 6 matchings, from which the controller can choose to play. To match the number of controller's actions, we also choose 6 random traffic matrices as the possible actions for the adversary. To make sure the system is stable, the 6 traffic matrices are all doubly-stochastic matrices with load $x < 1$.

### G.2 TRAINING ROBUST CONTROLLER

The overall training process proceeds as implied by our theoretical result. The controller, P1, and the adversary/nature, P2, take turns to pick an action: the controller schedules packets by picking one of the 6 matchings, $\sigma$, and then the state evolves to $\max(S - \sigma, 0)$; the adversary adds packets by selecting one of the 6 traffic matrices, $\lambda$. The number of arriving packets for each queue is then sampled according to a Bernoulli distribution with mean given by the corresponding entry in $\lambda$, and the state evolves to $(S' + \text{Ber}(\lambda))$; the above process continues. Each action, either from P1 or P2, is chosen by first running a finite-time MCTS and then playing the optimal action according to the resulting policy from MCTS. The policy network of each player is trained by supervised learning with cross-entropy loss. The optimal actions returned from MCTS serves as the training data.

**Policy and value networks.** We train two policy networks: one for the controller P1 and one for the adversary P2. Additionally, similar to AGZ, we also train a value network $V_\theta$ to help with MCTS, i.e., the optimal oracle is approximated by running a finite-time oracle that combines MCTS and a value network together. Note that this choice is largely motivated by the nature of the problem. Unlike AGZ, which is essentially an episodic game, the switch scheduling problem is a continuing task, where the system runs forever to schedule queued packets and receive new packets. The trees need to be very deep in order for the system to operate in stationary states. With the value network, one could quickly evaluate new nodes and update the statistics during tree search. This might speed up the MCTS process and hopefully, lead to a more efficient approximation for the optimal oracle. Note that only one value network is needed. In a zero-sum game, the value of the controller should be the negative value of the adversary. Finally, we remark that this overall setup indeed aligns with

our theory. Only the policy networks are trained to take actions: the value network is merely used to guide our MCTS search to generate training data for the policy networks.

**Training.** The networks are all feed-forward neural networks. Each neural network has 2 hidden layers with 24 ReLU units each. The input to these networks is a $3 \times 3$ state vector, i.e., the configuration of the 9 queues. Policy networks have a softmax layer at the end with 6 output units representing a probability distribution over the 6 possible actions. After each action, the networks are trained immediately (i.e., batch size = 1). The policy networks are trained to minimize the cross-entropy loss between its output (i.e., the predicted distribution over actions) and the policy obtained from MCTS. As elaborated before, the switch scheduling problem is posed as a continuing RL task. For such problems, minimizing the average reward is a more suitable measure. We train the value network using the differential semi-gradient SARSA (Sutton & Barto, 1998) algorithm. In particular, at each time after the controller receives a reward $r$ at a state $S$, the network is updated. We use Adam as our optimizer with a learning rate of $10^{-4}$.

**MCTS.** In each state that is either controlled by the scheduler or the adversary, an MCTS search is executed, guided by the value function $V_\theta$. The overall MCTS process is similar to the one used in AGZ. For each state controlled by the scheduler, it performs multiple iterations of MCTS. In each iteration, the basic search process uses a UCT (Kocsis et al., 2006) style approach to select the moves to traverse the tree. As in AGZ, once a new node is expanded, its value is evaluated using the value function $V_\theta$ and then propagated upwards to update the value estimates for all nodes on the search path. The path traversed during each iteration of the MCTS search tree alternates between the controller picking an action that maximizes its estimated average reward and the adversary minimizing it. At the end of the final iteration, the action from the root that has the maximum value is chosen to be the controller's action for the current step. Note that this action becomes the training data for the policy network of the controller. For each state controlled by the adversary P2, it also performs a similar MCTS, using $-V_\theta$ as its value function to guide the search, and selects the traffic matrix with the maximum value from its point of view in the end. For both the controller and the adversary, we perform 500 iterations of MCTS search at each step. The MCTS search tree is restricted to a maximum depth of 10.

