# OpenReview forum: "Understanding & Generalizing AlphaGo Zero"
_ICLR.cc/2019/Conference_

### Official Review · AnonReviewer2 · 2018-10-31
**heavy on notations, limited impact applicability / experimental results**

**Rating:** 5
**Confidence:** 3

**Review:**

The paper proposes a formal framework to claim that Alpha Zero might converges to a Nash equilibrium. The main theoretical result is that the reward difference between a pair of policy and the Nash policy is bounded by the expected KL of these policy on a state distribution sampled from the Nash policies.

The paper is quite heavy on notations and relatively light on experimental results. The main theoretical results is a bit remote from the case Alpha Zero is applied to. Indeed the bound is in 1/(1-/gamma) while Alpha Zero works with gamma = 1. Also

Casting a one player environment as a two player game in which nature plays the role of the second player makes the paper very heavy on notations.

In the experimental sections, the only comparison with RL types algorithm is with SARSA, it would be interesting to know how other RL algorithms, perhaps model free, would compare to this, i.e. is Alpha Zero actually necessary to solve this tasks?


---
p 1

' it uses the current policy network g_theta' : policy and value network.

p 2 / appendix
No need to provide pseudo code for alpha zero the original paper already describes that?

p2 (2). It seems a bit surprising to me that the state density rho does not depend upon pi but only on pi star?

p4:
Not sure why you need to introduce R(pi), isnt it just V_pi (s_0) ? Also usually the letter R is used for the return i.e. the sum of discounted reward without the expectation, so this notation is a bit confusing?

p5:
paragraph2: I don't quite see the point of this.

p8:
"~es, because at most on packet can get serviced from any input or output port.~" typo ?

---

> ### Author Response · Authors · 2018-11-26
> **Response to Reviewer 2**
>
> Thank you very much for your time reviewing our paper. It appears that certain main claims were misinterpreted, and we would like to take this opportunity to address the concerns:
>
> 1. Heavy on notations: the main concern in the review seems to be the notations. However, we would like to emphasize that this is a theoretical paper, aiming for a quantitative understanding of AlphaGo Zero, which is currently lacking. As you mentioned, we propose a formal framework to study and understand AGZ. To this end, certain level of mathematical rigor is absolutely necessary. We chose to use precise notations to make sure that our formal framework is mathematically correct and meaningful. Because the study was on two-play games, some statements do become more complex. However, these are unavoidable. Finally, we would like to point out that there are many literatures on games such as [1]  that do rely on ``heavy’’ notation to make precise mathematical claims.
>
> For reasons mentioned above, we are unable to understand exactly what the reviewer’s criticism precisely is about. Specifically, we would appreciate if the reviewer can provide feedback on what parts of the paper weren’t clear, as we believe it will help us improve the quality of the paper.
>
> 2. Casting MDP as a two player game: Here we were not deliberately trying to make the paper heavy on notation. The purpose is to show that this different viewpoint can lead to different ways of learning robust policies for agent. Without formally writing down the connection, we cannot show rigorously that our theorems extend to this robust MDP case. As mentioned in 1., this level of mathematical rigor is required for theoretical studies.
>
> 3. Gamma < 1: Most of the theoretical studies in RL study the infinite horizon problem with discounting (gamma < 1). This paper follows the same trend. Like many theoretical papers, this setting makes the formulation clean and analysis possible. Without discounting, the analysis for many algorithms is almost impossible or becomes very involved in notation. Actually, most of the successful algorithms were analyzed for this setting first, such as TRPO [2]. They may then be applied in practice for non-discounted settings. For example, although many games are finite-horizon in nature and rewards are not discounted, those algorithms can still be applied in practice successfully.
>
> 4. Experiments: We would like to emphasize that the main purpose of the paper is to provide a formal theoretical framework to analyze AGZ and to extend the algorithm to robust MDP problems. The value of those theoretical contributions should not be overlooked. We use the experiment to serve as a proof of concept that this formal framework can be applied to solve robust MDP problems. We are not attempting to claim that AGZ is the only method that can solve these tasks. Instead, the main message is that applying the AGZ robust MDP formulation is feasible in practice, and in such a  challenging problem, it achieves similar performances of the state-of-the-art algorithms (after decades of research).
>
> ----------
>
> Thank you for pointing out the typos. Here are additional comments related to other small concerns:
> p2/appendix:
> In fact, the original paper does not contain concise pseudo code. There are some concise discussions though. Even it does, we believe that giving pseudo code in our paper helps to set up the proper background and increase overall readability, without referring the readers to carefully read the original paper.
>
> p2(2):
> Note that, by definition, \pho is the state density under a particular policy. Therefore, \pho_pi^* is the state density visited under the optimal policy, and should not depend on pi at all. The more precise definition and derivation of the results are given in Section 3 and Appendix D, with additional discussions on the probability measure in Appendix F.
>
> p4:
> No, it is not V_\pi(s_0). Note that the initial state is distributed according to some initial distribution. R(\pi) is in fact the expected reward under $\pi$, i.e., expected value over V_\pi(s_0). We use letter R with the hope that this quantity is related to reward, and use \pi to clarify that this is the reward related to the policy. In fact, this type of notation does appear in other literature, eg. [3]. We could use other notation if it helps.
>
> P5:
> This is meant to precisely and formally introduce the mathematics of the robust MDP formulation. Please also refer to comment 2 above.
>
>
>
> [1] Perolat, Julien, et al. "Approximate dynamic programming for two-player zero-sum markov games." International Conference on Machine Learning (ICML 2015). 2015.
> [2] Schulman, John, et al. "Trust region policy optimization." International Conference on Machine Learning. 2015.
> [3] Pinto, Lerrel, et al. "Robust adversarial reinforcement learning." ICML(2017).

---

### Official Review · AnonReviewer3 · 2018-11-03
**Interesting insights about alphaGo Zero and a nice case-study.**

**Rating:** 7
**Confidence:** 4

**Review:**

This paper analyzes the AlphaGo Zero algorithm by showing that the optimal policy corresponds to a Nash equilibrium. The authors then show that the equilibrium corresponds to a KL-minimization. Finally, the show on a classical scheduling task.

On the positive side, the paper is well written and structured. The results presented are very interesting, specially showing that stochastic approximation of a KL-divergence minimization. The case-study is also interesting, although does not improve current state-of-the-art. On the negative side, I think the relevance and novelty of the results should be explained better.

For example, it is not clear the strong emphasis on the robust MDP formalization and the fact that MCTS finds a Nash equilibrium. The MDP formalization is rather straightforward. Also, MCTS has been used extensively to find Nash equilibria in both perfect and imperfect games, e.g., "Online monte carlo counterfactual regret minimization for search in imperfect information games". Maybe the authors can elaborate more on the significance/relevance of this contribution.

Besides, the power of AlphaGo Zero resides in the combination of the MCTS together with the compact representation learning of the value functions. The presented analysis seems to neglect the error term corresponding to the value function.

There are other minor details:

- Eq(2). notation: \forall s is missing
- Theorem 2 should be Theorem 1
- "there are constraints per which state can transition"
- "P1 is agent" -> "P1 is the agent"
- "Pinker" -> "Pinsker"
- C_R in Eq(5) is not introduced.

---

> ### Author Response · Authors · 2018-11-26
> **Response to Reviewer 3**
>
> Thank you for your encouraging comments. We agree with your suggestions and we will revise our paper accordingly. We will also comment on the gap between our analysis and AGZ in the introduction to make it clearer, and discuss potential future work (e.g., considering approximation errors due to MCTS and the value function) in the conclusion.

---

### Official Review · AnonReviewer1 · 2018-11-07
**The results in the paper are relatively straightforward and there is a clear gap.**

**Rating:** 5
**Confidence:** 5

**Review:**

This paper seeks to understand the AlphaGo Zero (AGZ) algorithm and extend the algorithm to regular sequential decision-making problems. Specifically, the paper answers three questions regarding AGZ: (i) What is the optimal policy that AGZ is trying to learn? (ii) Why is cross-entropy the right objective? (iii) How does AGZ extend to generic sequential decision-making problems? This paper shows that AGZ’s optimal policy is a Nash equilibrium, the KL divergence bounds distance to optimal reward, and the two-player zero-sum game could be applied to sequential decision making by introducing the concept of robust MDP. Overall the paper is well written. However, there are several concerns about this paper.

In fact, the key results obtained in this paper is that minimizing the KL-divergence between the parametric policy and the optimal policy (Nash equilibrium) (using SGD) will converge to the optimal policy. It is based on a bound (2), which states that when the KL-divergence between a policy and the optimal policy goes to zero then the return for the policy will approach that of the optimal policy. This bound is not so surprising because as long as certain regularity condition holds, the policies being close should lead to the returns being close. Therefore, it is an overclaim that the KL-divergence bound (2) provides an immediate justification for AGZ’s core learning algorithm. As mentioned earlier, the actual conclusion in Section 4.2 is that minimizing the KL-divergence between the parametric policy and the optimal policy by SGD will converge to the optimal policy, which is straightforward and is not what AlphaGo Zero does. This is because there is an important gap: the MCTS policy is not the same as the optimal policy. The effect of the imperfection in the target policy is not taken into account in the paper. A more interesting question to study is how this gap affect the iterative algorithm, and whether/how the error in the imperfect target policy accumulates/diminishes so that iteratively minimizing KL-divergence with imperfect \pi* (by MCTS) could still lead to optimal policy (Nash equilibrium).

Furthermore, the robust MDP view of the AGZ in sequential decision-making problems is not so impressive either. It is more or less like a reformulation of the AGZ setting in the MDP problem. And it is commonly known that two-player zero-sum game is closely related to minimax robust control. Therefore, it cannot be called as “generalizing AlphaGo Zero” as stated in the title of the paper.

---

> ### Author Response · Authors · 2018-11-26
> **Response to Reviewer 1**
>
> Thank you for the detailed comments.
>
> Our goal is to develop a quantitative understanding of AlphaGo Zero (AGZ), moving beyond the intuitive justification for the algorithms in the original work. We believe that a rigorous mathematical analysis is crucial to provide a solid foundation for understanding AGZ and similar algorithms. This requires developing (i) a precise mathematical model, (ii) a quantitative performance bound within the model.
>
> Our work takes an important step in this direction by modeling AGZ’s self-play and its supervised learning algorithm accurately. In particular, we use the turn-based game model to capture the self-play aspect. We develop a quantitative bound in terms of cross-entropy loss in supervised learning, which is the “metric” of choice in AGZ. While the cross-entropy loss seems intuitive, using it as a quantitative performance measure requires careful thought. For example, in Appendix F (page 19, 2nd paragraph), we discussed a scenario where this intuition is incorrect under a careless measure. That is, seemingly “obvious” algorithms can fail in the absence of a rigorous mathematical proof.
>
> We agree that there is a gap between AGZ and our model. As mentioned in our paper, MCTS converges to the optimal policy for both classical MDPs and stochastic games. Hence in this paper, we model the AGZ’s MCTS policy by the optimal policy, and mainly focus on the other two key ingredients of AGZ, self-play and supervised learning. It will be interesting to study how the error between MCTS and the optimal policy affects the iterative algorithm. This is a research direction we think is worth pursuing in the future.
>
> We also agree with the reviewer that some of our statements might be too strong. We will revise accordingly. Instead of ``immediate justification``, we believe this work does provide a first-step, formal framework towards a better theoretical understanding. We will also revise the title, perhaps to ``applying AGZ`` so as to make the connection to MDP more clear in our paper.

---

### Meta-Review · Area_Chair1 · 2018-12-17
**Contains interesting results, but certain key aspects were criticized and these criticisms were not addressed in the rebuttal.**

**Confidence:** 3
**Recommendation:** Reject

**Metareview:**

This work examines the AlphaGo Zero algorithm, a self-play reinforcement learning algorithm that has been shown to learn policies with superhuman performance on 2 player perfect information games.  The main result of the paper is that the policy learned by AGZ corresponds to a Nash equilibrium, that and that the cross-entropy minimization in the supervised learning-inspired part of the algorithm converges to this Nash equillibrium, proves a bound on the expected returns of two policies under the and introduces a "robust MDP" view of a 2 player zero-sum game played between the agent and nature.

R3 found the paper well-structured and the results presented therein interesting. R2 complained of overly heavy notation and questioned the applicability of the results, as well as the utility of the robust MDP perspective (though did raise their score following revisions).

The most detailed critique came from R1, who suggested that the bound on the convergence of returns of two policies as the KL divergence between their induced distributions decreases is unsurprising, that using it to argue for AGZ's convergence to the optimal policy ignores the effects introduced by the suboptimality of the MCTS policy (while really interesting part being understanding how AGZ deals with, and whether or not it closes, this gap), and that the "robust MDP" view is less novel than the authors claim based on the known relationships between 2 player zero-sum games and minimax robust control.

I find R1's complaints, in particular with respect to "robust MDPs" (a criticism which went completely unaddressed by the authors in their rebuttal), convincing enough that I would narrowly recommend rejection at this time, while also agreeing with R3 that this is an interesting subject and that the results within could serve as the bedrock for a stronger future paper.